# Identification of Follow-Up Markers for Rehabilitation Management in Patients with Vestibular Schwannoma

**DOI:** 10.3390/jcm12185947

**Published:** 2023-09-13

**Authors:** Frédéric Xavier, Emmanuelle Chouin, Brahim Tighilet, Jean-Pierre Lavieille, Christian Chabbert

**Affiliations:** 1Team Pathophysiology and Therapy of Vestibular Disorders, Laboratory of Cognitive Neurosciences, UMR7291, Aix Marseille University-CNRS, 13331 Marseille, France; brahim.tighilet@univ-amu.fr (B.T.); christian.chabbert@univ-amu.fr (C.C.); 2Unit GDR2074, CNRS, Research Group on Vestibular Pathophysiology, 13331 Marseille, France; manou44@hotmail.fr; 3Department of Otorhinolaryngology, Head and Neck Surgery, Hospital la Conception, Aix Marseille University, 13005 Marseille, France

**Keywords:** schwannoma vestibular, physiotherapy, instrumental indicators, follow-up markers

## Abstract

This study delves into the absence of prognostic or predictive markers to guide rehabilitation in patients afflicted with vestibular schwannomas. The objective is to analyze the reweighting of subjective and instrumental indicators following surgery, at 7 days and 1 month postoperatively. This retrospective cohort encompasses 32 patients who underwent unilateral vestibular schwannoma surgery at the Marseille University Hospital between 2014 and 2019. Variations in 54 indicators and their adherence to available norms are calculated. After 1 month, one-third of patients do not regain the norm for all indicators. However, the rates of variation unveil specific responses linked to a preoperative error signal, stemming from years of tumor adaptation. This adaptation is reflected in a postoperative visual or proprioceptive preference for certain patients. Further studies are needed to clarify error signals according to lesion types. The approach based on variations in normative indicators appears relevant for post-surgical monitoring and physiotherapy.

## 1. Introduction

The non-pharmacological management of vertiginous and unstable patients has undergone significant evolution through the adoption of evidence-based practices and the publication of validity criteria in this field [1,2]. The rehabilitative approach following peripheral vestibular impairment is based on studies detailing the alteration of vestibular function in cohorts without prior disorders [2,3,4,5,6]. However, epidemiology reports a non-negligible proportion of progressive peripheral impairments subject to compensatory processes that influence therapeutic application [7,8,9,10,11,12]. The outcomes of interventions in vestibular rehabilitation therapy (VRT) are thus more nuanced [2]. It appears that solely modeling static and dynamic syndromes is no longer sufficient to establish precise functional diagnosis or identify markers for rehabilitative follow-up. One of the main reasons is that the vestibular system involves other central structures [13] that play a role at different disease stages, giving rise to superimposed symptoms and syndromes of varying intensities.

Our study focuses on patients with vestibular schwannomas (VS), the most common benign tumors of the cranial fossa [14,15]. The incidence and prevalence increase with age, peaking between 50 and 60 years [16]. Magnetic resonance imaging (MRI) is the radiological reference for diagnosis and monitoring of VS [17]. The indication for conservative management through sequential surveillance or radiosurgery/radiotherapy are preferred choices for intracanalicular VS resection and radiosurgery/radiotherapy are the two therapeutic options proposed for tumors extending beyond the internal auditory canal (Koos Stage II and III or Tokyo Stage IIb and III). Consensus exists for optimal tumor excision surgery in compressive tumors (Koos and Tokyo Stage IV or diameter exceeding 30 mm [18]). Symptoms such as tinnitus, vertigo, and balance disorders have been studied in the context of conservative treatment. The association of vertigo with auditory symptoms appears to be a factor favoring physical and emotional decompensation [19]. However, in general, acute postoperative vertigo and chronic instability are consequences rather than complications of vestibular destruction or deafferentation. They exhibit a stable incidence and are not contingent on the improvement of surgical techniques. Preoperative criteria have been described to predict the risk of complications [20], including tumor size, level of hearing loss, and facial and neurological impairments. Yet, in our rehabilitative practice, we observe a chronicity that sets in after VS surgery in certain patients. The prevalence of this chronicity varies based on studies and populations, reporting rates of up to 30% of operated VS patients [21]. This implies that in certain patient profiles, vertigo can persist or appear chronically beyond 3 months post-surgical intervention. One of the identified causes is the presence of a significantly observable residual gain in near-total resections (more often than in total resections), which adversely impacts compensation prognosis [22]. This residual gain can be referred to as a subliminal post-surgical vestibular error signal (SEV). The absence of physical activity and a preoperative VS hyporeflexia in thermal examination are also two predictive factors of poor postoperative vestibular compensation [23]. To our knowledge, no study has presented a compilation of these evaluations or described the evolution of these indicators postoperatively to determine if there is (i) a particular kinetics of their reweighting that differs from the clinical model of acute unilateral vestibular hyporeflexia, well-described in both animal and human models for several decades [24,25,26], or (ii) a specific profile of the evolution of these indicators emerging from this analysis to explain the transition into chronicity. The originality of this study lies in the attempt to present the evolution of indicators over time, identify the reweighting of these indicators and their normalization, and deduce the neurophysiological consequences of vestibular compensation.

## 2. Materials and Methods

The tables describing the normative values are available in Appendix A.

### 2.1. Study Design

Our retrospective cohort study focusesn on the evaluation of follow-up paraclinical outcomes in patients with vestibular schwannomas (VS). Data collection took place between August 2019 and January 2020, in accordance with the guidelines of the competent ethical committee.

### 2.2. Study Population

The cohort studied included patients who consulted the Department of Otolaryngology and Cervico-Facial Surgery of the La Conception University Hospital in Marseille, France, between 1 January 2014 and 30 June 2019. A total of 155 patient records of patients diagnosed and operated on for unilateral vestibular schwannoma (VS) were selected (Figure 1). Of these, 66 patients were selected who met the following criteria: (i) Diagnostic criterion: patients with unilateral VS Tokyo 2003 IIb or III and absence of central damage; and (ii) Clinical follow-up criterion: presence of the inclusion and clinical follow-up sheet (including completion of at least 3 questionnaires, 3 vestibular assessments the day before surgery, 7 days after surgery, and 1 month after surgery, and at least one preoperative MRI report with tumor measurements). We carefully examined each medical record twice to detect any gaps or inconsistencies in the recorded information in order to avoid information bias. To minimize selection bias, we applied rigorous inclusion criteria based on specific characteristics of the patients and their vestibular tumors after re-reading the MRIs. After random selection, 32 records were selected, representing a significant proportion of the population concerned. In addition to minimizing subjective selection and ensuring equity from randomization, the collection of clinical data and the conduct of studies of 32 records were feasible in terms of resource availability. Indeed, (i) the health crisis imposed the administrative closure of services to research in science from January 2020; and (ii) the objective of the study is exploratory: the sample size is sufficient to obtain preliminary indications of the results and, if necessary, justify further studies.

### 2.3. Study Variables and Outcome Criteria

▪Tumor size and classification

The Tokyo classification, consisting of 6 grades proposed by Kanzaki in 2003, which is more appropriate for small-sized tumors as well as the interaction between size and hearing preservation, has been adopted for this study. The Tokyo classification considers both tumor size and its extension in various anatomical regions (notably the cerebellum), subdividing Koos Stage II into two stages: IIa and IIb. Only patients with tumors presenting as Stage IIb and III were included (Appendix A).

▪Hearing and classification in the vestibular schwannoma (VS)

For this study, the AAO-HNS classification (Appendix A) was preferred. This classification uses the word recognition score (WRS) or the maximum tolerated volume to assess auditory function. Variations in the contralateral side pure-tone average (PTA) from the day before the operation, the 7th day after the operation, and one month after the operation were included.

▪Vestibulometry test

Video-oculography (VOG) and Videonystagmography (VNG) were performed using the Dr. Hulmer’s VNG model by Synapsys-Marseille. Horizontal saccade tests were performed with VOG. The kinetic testing on the electronic chair with 0.25 Hz bursts was performed in 4 conditions, as well as the thermal nystagmography test (Appendix A).

Posturographic tests were conducted in 6 conditions according to Nashmer’s standards on the multitest platform model Framiral^®^. Data were recorded and analyzed based on the specified variables (Appendix A). Each condition was measured for 30 s, with a 15 s break between each condition. The area of the confidence ellipse containing 90% of all center of pressure positions was recorded and named the Center of Pressure Excursion Area (ECDP; in mm^2^). Energy consumption was measured in two axes (mediolateral: X and anteroposterior: Y) within three frequency bands, 0.05–0.5, 0.5–1.5, and 1.5–10 Hz through wavelet analysis using Posturopro^®^ Software v6.2.8, Inserm, Marseille, France. The following parameters were selected for the study: (i) Visual Dependency Index (VDI), calculated as the ratio of ECDP between [optokinetic stimulation on stable ground + optokinetic stimulation on unstable ground/eyes closed on stable ground + eyes closed on unstable ground] × 100. A VDI of 100% confirms the clinical state of visual dependence (VD). (ii) Postural Instability Index (PII), derived from wavelet and diffusion analysis based on the ratio between a power index (PI), representing the amount of energy the subject expends to maintain balance on the platform for a specific sequence, and a postural control index (PCI), representing the total duration during which no energy consumption was measured (calculated by Posturopro^®^ Software v6.2.8: PII = PI/PCI). A lower PII indicates better subject postural control and vice versa. (iii) Presence or absence of falls during the examination.

The subjective visual vertical (SVV) was assessed using the Frami-VS equipment by Framiral^®^, resulting in a calculated mean value. For each trial, the line is presented in a balanced pseudo-random sequence between left and right, with 10 trials per side. Conventionally, in normal subjects, a negative sign indicates a leftward tilt of SVV, and a positive sign indicates a rightward tilt of SVV (Appendix A).

▪Self- or hetero-evaluative questionnaires

The Dizziness Handicap Inventory (DHI), the Penn Acoustic Neuroma Quality-of-Life (PANQOL), and the Short Form (36) health survey (SF 36) were employed to assess handicap, SV-specific quality of life, and overall health status, respectively (Appendix A).

### 2.4. Data Collection and Statistical Analysis

A preliminary flat sorting in Excel was conducted to perform a double-check review for data entry errors, as well as to manage the recording and handling of missing values using simple imputation. We ensured the consistency of collected data from various sources to minimize errors and measurement biases. A cross-tabulation streamlined the data, organizing them based on clinical scales and distributions by dimensions for each questionnaire. Data from instrumental tests were analyzed against available norms when applicable. A univariate and multivariate intention-to-treat analysis was also conducted using R software version 4.2.3 and the R Studio interface. In total, 54 variables were studied across 3 time periods: T1, T2, and T3, corresponding respectively to the day before surgery, the 7th day after surgery, and one month after surgery. The analysis proceeded through multiple stages:▪Analysis of Rate of Change

For the study of the rate of change, an analysis based on data availability was performed (available case analysis). When the starting or ending value is 0, the rate of change is not calculated, as this would result in a rate of change of infinity. Similarly, if there are too many missing values, the rate of change is not calculated. Uncalculated rates are shown as blank cells in the tables. For simplified reading, only statistically significant results are presented to the reader. The complete data set is available upon request. The % change = ((Final Value − Initial Value)/Initial Value) × 100.

Once the null hypotheses (H0: no significant differences) and alternative hypotheses (H1: significant differences) are defined, the statistical treatment depends on the indicator being compared and whether or not it follows a Gaussian distribution. The normality test used is the Shapiro–Wilk test. H0 is rejected at a 5% significance level. If the distribution is normal, the test used for difference is the paired Student’s *t*-test. If the distribution is not normal, the test used for difference is the sign test. It is important to note that the McNemar’s chi-squared test was used to measure the difference in variation specifically for the indicator “Postural Fall” due to the qualitative/categorical nature of this indicator (Yes/No).

Interpretation of results: if *p* ≤ 0.05, we reject H0 and conclude that there is a significant difference between initial and final values (not due to chance). It should be noted that the larger the sample size, the more likely statistically significant differences are to be detected, even if these differences are minimal in practical terms. Therefore, the chosen sample appears suitable for clinical interpretation.

▪An analysis using Cramer’s coefficient

This is a measure of association between two nominal or categorical variables, C = √ {(X^2^/n min (k − 1, r − 1)}. It is used to assess the strength of the relationship between these variables and ranges between 0 and 1. The closer the value of the Cramer’s coefficient is to 1, the stronger the association between the variables, while a value closer to 0 indicates a weaker association.

## 3. Results

The tables presenting the results of the statistical analysis are available in Appendix A.

### 3.1. Sample Description

The sample consists of 41% males and 59% females, with a mean preoperative age of 59 years (Table 1). The mean age of the patients at baseline was 59 years, with a standard deviation of 11 years. The mean duration of preoperative follow-up was 3.8 years, with a standard deviation of 6.7 years.

### 3.2. Study of Indicator Evolution

A simplified representation of the results is provided in the form of color-coded tables. These tables respectively describe indicators related to: posture, eye movements, kinetic analysis, thermal analysis, and questionnaire dimensions for the entire study group.

#### 3.2.1. Evolution of Indicators from Postural Analysis

Over the interval T1–T3 (Table 2), four out of nine indicators show no statistically significant fluctuations: ‘VVS’; ‘unstable EO PII’, ‘fall’, and ‘stable EO PII’. The others show clinical improvement. During the T1–T2 period, a significant deterioration in the ‘VVS’ indicator and the ‘unstable EO PII’ indicator is observed. In T2, the percentage of fall-prone patients is 87.5% across all conditions, compared to 43.8% in T1. The ‘stable EO PII’ indicator does not fluctuate significantly in this period, unlike the ‘stable EC PII’, ‘stable OKN PII’, and ‘IDV’ indicators. The two unstable conditions, EC and OKN, could not be performed during this period. Finally, over the T2–T3 period, seven indicators show positive evolution, meaning clinical improvement is observable. The percentage of fall-prone patients in T3 is 34.38%. Only two indicators do not show a significant rate of variation during this period: the ‘stable EO PII’ indicator and the ‘unstable EC PII’ indicator.

#### 3.2.2. Evolution of Indicators from Video-Oculography (VOG) Analysis

During T1–T3, VOG indicators do not show significant fluctuations in their rates (Table 3). In T1–T2, saccades show a clinically significant deterioration in terms of speed and precision, while latency improves. Finally, in T2–T3, indicators of speed and precision show a clinically favorable and statistically significant improvement. During this period, latency does not fluctuate significantly.

#### 3.2.3. Evolution of Indicators Derived from Kinetic Videonystagmography (VNG_k_)

In Table 4, over the T1–T3 period, only two indicators show clinically unfavorable fluctuations: ‘VOR2 gain’ and ‘VVOR prep’. In T1–T2, two indicators do not exhibit statistically significant rates of variation: ‘IFO gain’ and ‘IFO prep’. Lastly, in T2–T3, only the indicators of preponderance in ‘VVOR’, ‘VOR’, and ‘VOR2’ demonstrate significant clinical improvement.

#### 3.2.4. Evolution of Indicators Derived from Thermal Videonystagmography (VNG_t_)

Table 5 shows that in T1–T3, only the rate of variation of the ‘VNG_t_ deficit’ indicator is statistically significant and clinically unfavorable. In T1–T2 and T2–T3, the variations of all three studied indicators are statistically significant. In T1–T2, the variation is clinically unfavorable, while in T2–T3, it is clinically favorable.

#### 3.2.5. Evolution of Indicators from the Analysis Using Questionnaire Dimensions

In T1–T3 (Table 6), the ‘emotional well-being’, ‘fatigue/energy’, and ‘physical limitation’ dimensions of the SF-36 questionnaire show favorable evolution, while the ‘physical’ dimension of the DHI questionnaire deteriorates. In T1–T2, three indicators exhibit clinically favorable evolution: those from the ‘emotional well-being’, ‘physical limitation’, and ‘general health’ dimensions of the SF-36 questionnaire. Three indicators have a clinically unfavorable evolution during this period: that of the ‘hearing’ dimension of the Panqol questionnaire, as well as the ‘physical’ and ‘functional’ dimensions of the DHI questionnaire. Lastly, in T2–T3, four indicators improve: the ‘emotional well-being’ and ‘fatigue/energy’ dimensions of the SF-36 questionnaire, as well as the ‘physical’ and ‘emotional’ dimensions of the DHI questionnaire.

#### 3.2.6. Study of Tonal Hearing Loss (PTA) Variation

The evolution of PTA is very pronounced and significant over the entire period (T1–T3) on the pathological side. The majority of this change occurs specifically between T1 and T2. In contrast, the evolution of PTA on the healthy side during T1–T3 and T1–T2 does not reveal a clinically significant change, but the period T2–T3 indicates a significant clinical improvement (Table 7).

### 3.3. Study of Normative Variables

#### 3.3.1. Study of Posturographic Indicators

The study of posturographic indicators reveals a lack of normality in T1 for stable eyes-open (EO) PII indicator in 40.6% (n = 13) of patients, stable eyes-closed (EC) PII indicator in 31.3% (n = 10) of patients, and stable optokinetic nystagmus (OKN) PII indicator in 28.1% (n = 9) of patients (Figure 2). The evolution of the PII indicator in T3 is within the norms for the stable EO condition (87.5% (n 12 + 16) of patients), stable EC (71.9% (n = 23) of patients), and stable OKN (75% (n = 24) of patients). A postoperative effect is observed in the T1–T2 interval for all three indicators, with the most pronounced effect seen in the stable EC PII indicator (n = 14).

More specifically, the study of the visual dependence index shows that it is positive for 65.6% (n = 21) of patients in T1, 75% (n = 24) of patients in T2, and 34.4% (n = 11) of patients in T3. For this indicator, we assessed the transition to norms for the three intervals T1–T3, T1–T2, and T2–T3 (Figure 3). Thus, we have a total of 24 subjects exhibiting visual dependence during the T1–T2 period, compared to 11 during the T2–T3 period and 12 over the entire follow-up duration. In the T1–T2 interval, an unfavorable postoperative effect is observed in 25% of patients (n = 16), while during the T2–T3 interval, 50% of the cohort normalize their visual dependence index.

The evolution of normal VVS conditions in the sample is described in Figure 4 for the three time periods. A total of 56.3% (n = 18) of the subjects exhibit normal VVS at T1, compared to 46.9% (n = 15) at T3, and only 9.4% (n = 3) at T2. Following the intervention, an unfavorable postoperative effect is observed between T1 and T2 in 50% (n = 16) of the patients, while 35.5% (n = 12) of the subjects show favorable progression between T2 and T3.

#### 3.3.2. Study of Saccade Analysis Indicators in Video-Oculography (VOG)

We examined the evolution of the number of patients within norms for each of the three variables (latency, velocity, and accuracy) during the T1–T3 period (Figure 5). At T3, the percentage of subjects with right latency below the norm is 78.1% (n = 28) compared to 62.5% (n = 20) for left latency. During the same period, the number of patients achieving normative values for velocity and accuracy is 81.2% (n = 26) for right velocity, 71.9% (n = 23) for left velocity, 90.6% (n = 29) for right accuracy, and 96.9% (n = 31) for left accuracy.

#### 3.3.3. Study of Kinetic Indicators

The assessment of the four variables derived from burst mode kinetic tests under the conditions of visual-vestibulo-ocular reflex (VVOR) sensitivity, vestibulo-ocular reflex (VOR), dual-task vestibulo-ocular reflex (VOR2), and ocular fixation index (IFO), allowed us to understand the evolution of normalization or persistence outside the norms of these variables from T1 to T3 (Figure 6).

The norm for an individual was met by meeting both conditions of normality (on gain and preponderance). At T3, the percentage of patients within the norms is 65.6% for IFO, 44.0% for VVOR, 21.9% for VOR, and 18.8% for VOR2. A minority transitioned from a non-compliant situation to a compliant normative situation. Permutation analysis of an indicator to norms based on another was examined using the Cramer coefficient. It stands at 0.73 (*p* < 0.001) between VOR and VOR2 (Appendix A). In the multivariate analysis, it appears that the trajectories of the four variables do not evolve jointly. These results do not seem to corroborate the hypothesis that achieving norms for one variable is also associated with achieving norms for the other variables.

#### 3.3.4. Study of Indicators in Thermal Videonystagmography (VNG_t_)

The evolution of compliance with norms (Figure 7, Appendix A) reveals that for reflectivity, 40% of subjects who were within norms at T1 are no longer within norms at T3, and for preponderance, 35% of patients within norms at T1 are no longer within norms at T3. Ipsilateral deficit is acquired for the entire population under study.

## 4. Discussion

### 4.1. Description of the Sample

In line with the literature, the distribution of age and sex in our sample is representative of the population generally studied in this field [27,28,29]. The comorbidities reported are also similar to other studies [30], and they vary depending on the geographic area and ethnicity. While studies on vestibular schwannoma (VS) rightly focus on comorbidities that can impact peri-operative outcomes, this study focused on the short-term consequences of surgical treatment of patients with VS stage IIb and III according to the Tokyo classification, by examining a set of indicators. According to the literature, neither the operated side nor the surgical approach influences vestibular compensation [31]. The physiotherapy treatment applied to 93.8% before surgery and 100% after surgery homogenizes the population. While the management in France remains heterogeneous in this area, which could represent a confounding bias, this peculiarity reflects a field reality [32].

### 4.2. Study of the Evolution of Indicators

The behavior of an indicator can be assessed through different analytical approaches, two of which consist of following the significant evolution in its rate of variation over different periods and determining whether it meets a state of compliance with established norms over these same periods. The two approaches provide valuable information on the behavior of an indicator and can be used together to provide a complete understanding of the kinetic evolution of a subject who has undergone an SV operation.

As part of the evaluation of subjects who have undergone surgery for SV, a multitude of follow-up indicators were collected, encompassing objective, instrumental, and subjective aspects from questionnaires. Among these indicators, we selected those that best reflect the typical activity of physiotherapists in private practice in France [32], a crucial step in understanding current practice.

Overall, by focusing exclusively on the factors inducing normality, we note that nearly one-third of the population studied is unable to comply with the norms one month after surgery. This observation resonates with the observation introduced at the beginning of the study, where it was found that 30% of patients develop chronicity after surgical treatment for SV. From this perspective, the approach based on the rate of variation provides information on the kinetics of re-weighing of the indicators, which seems complementary and sometimes more relevant.

▪Postural indicators (Table 2)

The study of the rate of variation of postural indicators in this cohort suggests a specific recovery kinetics:⇨The Postural Instability Index (PII) in the stable eyes-open condition is not a significant follow-up indicator during the study periods. The study of the norms shows that more than 75% of the patients in the study remain stable. This finding makes it a robust indicator: a static balance defect maintained with eyes open during this period should be questioned and brought to the attention of the surgeon.⇨A late-recovery kinetics is observable, in accordance with the literature [23,33,34], of the indicators in the following conditions: stable eyes-closed, stable under optokinetic stimulation, unstable eyes-open, and unstable under optokinetic stimulation. However, the evolution of the rate of variation of the unstable eyes-closed PII indicator is not significant for the T2–T3 period. The study of the transitions to the norms in the T1–T2 interval shows that the PII in the stable eyes-closed condition is mainly degraded in this period, while the management of visual conflict seems to improve early by the transition to the norms of the postural instability indicator in stable condition under optokinetic stimulation (PII stable OKN; Figure 2). In conclusion, for physiotherapists not equipped with posturography equipment, progress in the conditions of balance with eyes closed and under optokinetic stimulation should be identified and sought in the static and dynamic conditions in the first month after surgery. If the Modified Clinical Test of Sensory Interaction on Balance (CTSIB m [35]) is practiced, then the progression should focus on conditions 2 and 4. A lack of progress should be questioned.⇨The assessment of the visual dependence index (IDV) is linked to the calculation mode of the equipment used and the type of visual stimulation to which the patient is subjected (optokinetic, slave vision). In our study, the calculation of the IDV shows a favorable and significant clinical evolution from the T2–T3 interval. A postoperative increase in visual dependence (DV) is noted, with a late normalization of this index for half of the group (Figure 3). Among the 21 DVs identified preoperatively, 11 were resolved after one month. This contrasts with the results obtained by Deveze et al. [36] in a cohort of patients with Meniere’s disease treated with VIII-nerve neurotomy, as well as by Parietti et al. [37] during the follow-up of a cohort of patients with SV treated by surgery. Excluding differences in the value and instrumental calculation of visual dependence, which vary according to studies and equipment used, these two publications are mainly distinguished by the composition of their respective populations. The first study presents a different population, while the second presents a less-homogeneous selection in terms of Koos classification (grades 1, 2, and 3). In our study, we assume that the homogeneity of our population in terms of tumor size and a minimum follow-up of three years are factors that could influence our results.⇨Immediately after surgery, two evolutionary parameters are well-identified: the immediate deterioration of dynamic open-eye balance and the occurrence of falls. In parallel, a significant worsening of subjective visual vertical (SVV) also illustrates this evolution [36]. In our cohort, the immediate postoperative effect has a deleterious effect on the SVV, characterized by a notable variation rate in T1–T2 and 90.6% of patients in the cohort not meeting the norms at 7 days postoperative (Figure 4). However, the response kinetics manifest themselves in the second period.


▪Indicators derived from video-oculography (VOG; Table 3, Figure 5)


In this study, several observations emerge regarding the evolution of saccade indicators following SV surgery. First, the analysis of compliance with norms for saccade speed and precision reveals that the majority of subjects normalized one month after surgery, although the analysis of variation rates highlights a temporary degradation before an improvement in T2–T3. In contrast, the analysis of saccade latency indicates that the majority of subjects did not meet norms before or after surgery; however, postoperative improvement emerges from the variation rates, although the indicator does not yet reach normality.

▪Indicators derived from the kinetic videonystagmography test (VNG_k_; Table 4, Figure 6)

The first remarkable point is the stability of the ocular fixation index (IFO) in the kinetic test. Indeed, even if in T3 34.4% of patients are not within the norms, the variation rate shows that its fluctuation is not significant. This makes it, like the stable PII indicator with eyes open, a robust indicator for the follow-up of patients of the same type; any modification of this index must be discussed with the surgeon.

Regarding the variations of preponderance indicators in burst kinetic tests, sensitizing the vestibulo-visual-ocular reflex (VVOR) (eyes-open study without fixation), the vestibulo-ocular reflex (VOR) and the VOR2 (VOR in dual task by mental calculation) show a postoperative aggravation that is consistent with the literature in the case of the VIII nerve section by neurotomy [38]. However, the particularity of this cohort lies in the significant increase in VOR gain, while that of VVOR and VOR2 drops for the T1–T2 interval. This two-stage kinetics of immediate degradation and delayed progression corresponds to the postoperative adaptation period necessary for the decrease in spontaneous nystagmus [35]. Indeed, functional recovery of the VOR can only occur after the reweighting of peripheral synaptic connections and the modulation of central compensation. The analysis of compliance of these indicators (normality gain + preponderance) segregates the cohort such that one-third of the subjects were not within norms at T1 and are still not at T3, and one-third of the subjects lose compliance with these indicators one month after surgery, while a significant progression begins in T2–T3 for the entire group.

For the physiotherapist, we have demonstrated that a single indicator does not allow us to quantify kinetic recovery. We considered the norm of these indicators based on gain and preponderance for each of them. However, in clinical practice, gain and preponderance evolve differently for each patient, depending on the different nosologies for the same indicator [39], which makes the VNGc a complex examination to understand. Nevertheless, their improvements should be sought as soon as spontaneous nystagmus disappears and during the first month after surgery. Their evolutions must be confronted with subjective evaluations, but their systematic normalization must be discussed. Our study does not allow us to identify the evolution of these kinetic markers in the norm over a long period, but our observation of the two-thirds of the cohort that is not within norms for VVOR, VOR and VOR2 over a period of one month could suggest the existence of a plateau effect, inducing specific normal thresholds for this patient profile.

▪Indicators from the thermal videonystagmography (VNGt) test; Table 5, Figure 7)

The entire cohort shows a deficit in the thermal test that no longer meets the norms at T3, a characteristic trend in this context. The delayed recovery of a deficit, as is the case in T2–T3 for this indicator, even below the norms, could indicate partial deafferentation. When correlated with the assessment of residual gain in postoperative VHIT, it can serve as a significant indicator of this major complication of perceptual-motor reweighting in this group of patients [22]. Similarly, a preserved preoperative caloric function has an impact on the catch-up saccades observed in postoperative VHIT, identified by the poorer performance of these [40]. However, a preoperative deficit is correlated with poor compensation of balance function in the postoperative period [23]. The kinetics of variation of the reflexivity and absolute preponderance is typical of the consequence of the VIII nerve shunting by neurotomy. On the other hand, the fact that nearly half of the cohort does not meet the norms at T3 on these two indicators should question the type of compensation strategy employed by the patient and a potential plateau effect of compensation [41]. On this subject, a renewed interest has arisen in the post-caloric recruitment index (PCRI). This parameter reflects the way in which the vestibular nuclei compensate for peripheral lesions, and indicates the level of activity of the vestibular pathways after thermal stimulation [42]. PCRI is particularly useful for assessing asymmetry between the two sides of the vestibular system and for predicting the stage of central compensation after a lesion. It is a parameter for physiotherapists to consider, and could allow the identification of the state of compensation and its threshold.

▪Subjective indicator (Table 6)

Three questionnaires were submitted in this study. We chose to analyze not the total scores but the scores of the dimensions present for each questionnaire. The French version of the Panqol questionnaire is by far the least relevant for the follow-up of our cohort. For the SF-36 questionnaire, the dimensions of “physical limitation”, “fatigue/energy”, and “emotional well-being” are good indicators of follow-up in this study. We note the statistically significant variation of the dimensions of “physical limitation” and “general health” in a favorable way, while the dimensions of “physical” and “functional” of the DHI worsen for the T1–T2 period. The explanation lies in the specificity of the dimensions involved. The SF-36 is a general questionnaire: it assesses physical capabilities in terms of activity limitation (walking distance, climbing stairs, etc.). The DHI, on the other hand, remains a specific tool for vestibular vertigo, as demonstrated recently in the literature [43]. The physical dimension of the DHI assesses physical activities that potentially induce the symptom of vertigo or instability. Its statistically significant variability at the three intervals studied makes it the most robust tool for tracking subjective symptoms in the study.

▪PTA Indicator (Table 7)

Pure-tone audiometry (PTA) should be analyzed in two stages. The PTA on the pathological side is a sensitive indicator of the surgical effect, as is the AAO-HNS scale, explaining the high rate of significant variation in T1–T2. The anatomical lesion of the translabyrinthine (TL) approach does not allow for any further variation in the second period; only complications such as post-surgical inflammatory consequences have an impact on the T2-T3 period for the retrosigmoid (RS) approach. However, the PTA on the healthy side shows a significant improvement in the T2–T3 period (−5.2%), which is new in the observation of sensory responses in these patients. This implies a modulation of the contralateral side, the central origin of which seems relevant. The results obtained by Heinrich [44] show that an excess of cognitive load can modulate auditory sensitivity, as assessed using (PTA). Other studies show that reduced cognitive abilities (which can be linked to fatigue) impact the degree of hearing loss [45,46]. All of these elements support the hypothesis that cognitive load influences post-surgical compensation in patients with SV.

In conclusion, this section shows several subtle indicators of an unexpected effect in the immediate postoperative period, notably in the improvement in saccade latency, gain in VOR condition, and the “emotional well-being”, “physical limitation”, and “general health” dimensions of the SF36.

### 4.3. Vestibular Schwannoma and Error Signal

The origin of the vestibular error signal is the disruption caused by vestibular schwannomas (VS) in the normal function of the vestibular nerve. The VS, which arise from Schwann cells that wrap the myelin sheath of the vestibular nerve, induce both histological and electrophysiological changes that disrupt signal transmission [47,48,49,50,51]. The effects of these changes alter the detection and discrimination thresholds [52], which may explain the loss of vestibular signal. Additionally, these studies suggest that VS can increase the level of neuronal noise, introducing unwanted interference into the true signal. This contributes to a decrease in the accuracy of signal transmission, which could impair the system’s ability to distinguish signals precisely [52]. The elevation of detection thresholds and a change in signal sharpness strongly suggest that VS induces a vestibular error signal (VES). The VES causes a reorganization of neural networks over time, beginning with the first structural lesions, thanks to the phenomenon of plasticity. The adaptation of the central nervous system to a progressive lesion is a complex process that involves the reconfiguration of neuronal connections, the adjustment of signal processing mechanisms, and the strengthening of remaining signaling pathways.

Analysis of the variations in tracking indicators, beyond their normality, made it possible to identify specific mechanisms, corroborating the VES in SV patients. In fact, the vestibular exploration condition during sinusoidal tests on the rotating chair showed that the gains of the VOR2 and the VVOR decrease, while that of the VOR increases. However, studies based on normative VOR states show a degradation of this indicator after unilateral VIII nerve deafferentation [36,38]. This adaptation suggests neuromodulation mechanisms around VOR put in place before the operation and incompletely repondered 7 days after the operation. Indeed, although the suppression of the VES by surgery can have beneficial effects on the processing of the vestibular signal from the healthy side, the strengthening of the intact pathways, the reconfiguration of brain plasticity and the reduction of interference, this does not imply the complete elimination of the compensation mechanisms put in place for several years. Nevertheless, depending on the level of initial adaptation, the postoperative reponderation will be different, as shown by the data of this study.

The literature specifies in this sense that the preoperative VOR adaptations can influence its accuracy (gain) or its precision (time constant) [52]. By modifying the accuracy of the VOR, the system seeks to improve the correspondence between eye movements and head movements to maintain stable and clear vision. Adjusting the precision of the VOR aims to reduce coordination errors between eye movements and head movements to reduce the sensation of dizziness and instability. Previous research specifies [39] that unilateral loss results in asymmetries in the gain and time constant of the VOR, independently of each other. Adjustment mechanisms aim to restore an appropriate balance between these two parameters. They are associated with changes in the resting rates of vestibular neurons (VNs) as well as adjustments in the projection forces in the commissural network of bilateral VNs. These adjustments contribute partially to restoring the gain of the VOR and eliminating the bias induced by the lesion. Other evidence suggests that the central nervous system (CNS) compensates for vestibular deficits by integrating oculomotor velocity parameters. The phase and gain of the VOR can be modified by the activity of neurons coding head movement and also those coding eye velocity (oculomotor proprioception). Therefore, eye velocity transmitted by oculomotor proprioception contributes to VOR learning. This process is thought to be subject to plasticity in the cerebellum [53]. These arguments suggest that patients who develop a proprioceptive preference in response to vestibular error signal (VES) acquire an increased sensitivity to movement in the VOR condition. When attention is directed towards a visual input (VVOR) or a cognitive task (VOR2), there is inhibition of the proprioceptive preference [54] and the gains in these two conditions are affected (Table 4). This state of proprioceptive preference seems to be both an adaptive mechanism in response to VES before surgery and also an established preference in postoperative compensation, as illustrated by the significant improvement in posturography results in stable and unstable conditions under optokinetic stimulation in T2–T3. These conditions express the performance of the subjects in the recruitment of this modality. Although these results suggest a stereotyped sensory reweighting, the study of IDV and DV nuances this finding. Indeed, half of the cohort did not resolve their DV after one month. However, we know that DV is associated with a vestibular compensation deficit when it is excessive [55]. This suggests that perceptual-motor reweighting cannot be reduced to visual and proprioceptive preferences [37]. Several arguments show (i) that the reweighting kinetics is heterogeneous in our results, and (ii) that the pathways that transmit sensory signals are sometimes identical and that their integration takes place on convergence areas in the CNS, so that the categorization of a subject according to their type of compensation is not so obvious [56].

Indeed, several authors suggest that sensory adaptations are not limited to the vestibular system itself, but affect the interactions between the high-level sensorimotor-perceptual systems integrating audition, somatosensation, cognition, etc. [53,57,58,59]. The favorable evolution of the variation rate for pure-tone audiometry (PTA) on the healthy side in the second period confirms the impact of vestibular compensation on global cognitive functioning. A very rich literature [60,61,62,63,64,65,66,67,68] indicates that audiovestibular losses can influence the processing of sensory inputs by reducing cognitive resources for other tasks while generating cognitive overload. These findings could also explain the delayed compensation of patients or the failure to optimize the sensory weighting after SV surgery for some indicators. In addition, the improvements observed in the subjective indicators of well-being and general physical condition of the SF 36 questionnaire could be the reflection of the consequences of prolonged SEV on the overall health of patients, given the cognitive load required to constantly compensate for SEV. The overall set of these results highlights the complexity of the mechanisms of compensation and plasticity and opens promising perspectives for understanding sensory interactions in vestibular pathologies.

Same findings apply to the study of voluntary eye movements related to the disruption of signals transmitted by the vestibular nerve in the presence of the tumor [47,51]. The close connection between the vestibular and visual systems suggests an adaptation of oculomotor responses to these vestibular error signals, which in turn permanently alter oculomotor pathways through plasticity mechanisms. Surprisingly, the expected clinical worsening of oculomotor function after surgery does not affect saccade latency. The complex neural circuits of latency, compared to those of speed and accuracy, could explain these differences in evolution, reinforced by postoperative mechanisms aimed at reducing spontaneous nystagmus. If such an improvement occurs in latency but not in speed or accuracy, it implies a reorganization of the latency circuits well before surgery. Indeed, the execution of voluntary saccadic eye movements involves the sensory detection of a peripheral target, sensory-motor integration, and motor execution. Therefore, the initiation of a saccade (latency) can be influenced due to a disrupted integrative processing [69,70]. Our observations indicate that the first postoperative days reflect a partially compensated oculomotor reorganization 7 days after surgery. These results therefore suggest a dynamic relationship between the reorganization of oculomotor pathways, visuo-vestibular compensation, and the underlying neuronal mechanisms. The increase in saccade latency would be one of the mechanisms of adaptation to SEV, as is the gain in VOR. Finally, the persistence of fine visual disturbances such as blurred vision, difficulty tracking moving objects, fixation or visual-tracking problems in situations such as driving, as well as visual fatigue, especially during reading, could be due to subtle perturbations of the oculomotor system induced by changes in precision, latency and speed of non-compensated postoperative saccades. The persistence of these symptoms should guide physiotherapists in monitoring the evolution of these indicators rather than in a simple assessment of their compliance. If necessary, an orientation towards corrective or facilitating visuo–visual signal treatments should be considered during vestibular rehabilitation therapy (VRT).

### 4.4. Study Biases

The sample size may potentially contribute to the lack of significant difference within the T1–T3 interval. Although the differences between T1–T2 and T2–T3 are substantial, detecting a significant difference between T1 and T3 could be challenging, due to the fewer subjects available to detect such a difference. Thus, all presented significance values (*p*-values) can be interpreted with limited risk at both extremes (*p* < 0.01 and *p* > 0.5). However, values fluctuating between 0.1 and 0.3 should be investigated in a larger cohort (gray-shaded portion of the tables).

## 5. Conclusions

Our study addresses the concept of temporal kinetics of reweighting follow-up markers, allowing us to discern two types of central nervous system behaviors: preoperative adaptation and postoperative compensation. It appears that the critical 7-day period following surgical intervention becomes a stage for competition between adaptive residues and compensation initiated by the deafferentation of the VIII nerve. Several indicators suggest that the physiotherapist should adopt a fresh approach during this crucial period to prevent the shift towards chronicity. This might involve (i) addressing residual visual or proprioceptive preference from preoperative adaptation, (ii) assessing the preoperative compensation level, particularly through thermal deficit analysis and visual dependence, (iii) identifying a postoperative residual signal through variations in thermal VNG deficits and gains in video head impulse testing, and (iv) managing postoperative fatigue, monitoring persistent altered emotional states, and accounting for fluctuations in other sensory indicators such as hearing. All these markers of cognitive difficulty mirror the alteration of sensory–perceptual–motor reweighting.

Observing these follow-up markers should prompt VRT to evaluate the patient’s overall cognitive resources (cognitive load, mental fatigue, and emotional state) before initiating vestibular rehabilitation therapy (VRT), as an ill-suited VRT approach can quickly prove ineffective for certain profiles. Given the highly specific sample of the study, extrapolating these initial results to a population with differing initial pathologies would be unwarranted, as the same follow-up markers may exhibit varying sensitivities to change. It would be prudent to consider profiling marker kinetics based on etiology. A thorough knowledge of established norms, where available, by the physiotherapist appears to be a cornerstone in this endeavor.

## Figures and Tables

**Figure 1 jcm-12-05947-f001:**
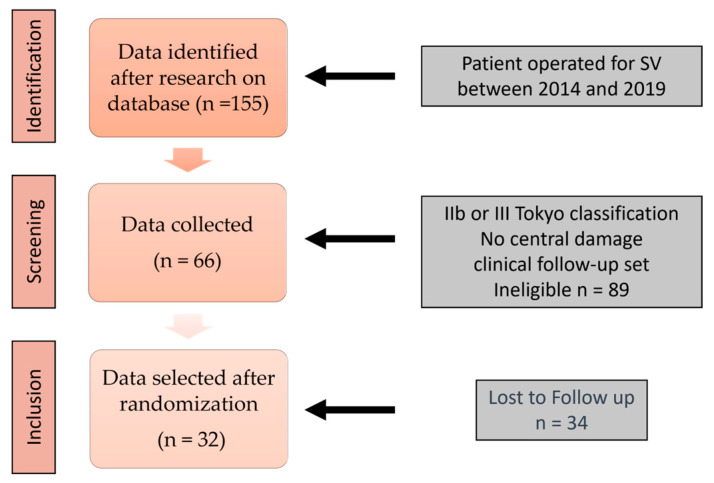
Flow diagram of inclusion of patients with bilateral vestibular hypofunction and distribution of diagnoses. n: number of patients.

**Figure 2 jcm-12-05947-f002:**
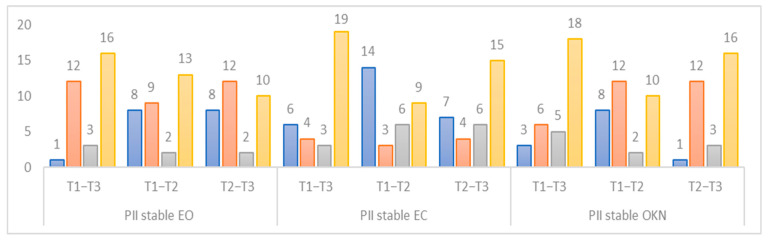
Change in normative status of the PII indicators for three periods studied. Blue: not up-to-standard for all periods studied. Orange: not up-to-standard at the beginning but up-to-standard at the end of the period studied. Grey: starts as standard at the beginning but not standard at the end of the period studied. Yellow: up-to-standard for all periods studied.

**Figure 3 jcm-12-05947-f003:**
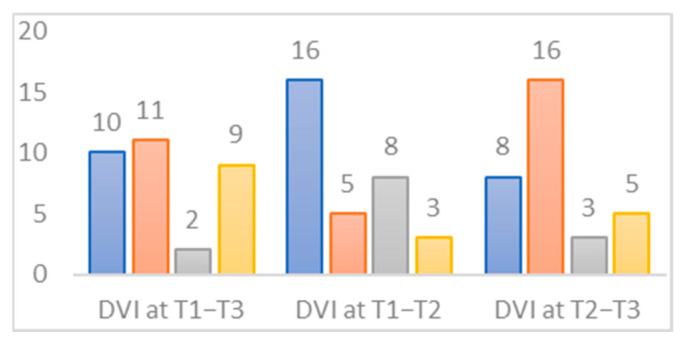
Change in normative status of the IDV indicator for three periods studied. Blue: not up-to-standard for all periods studied. Orange: not up-to-standard at the beginning but up-to-standard at the end of the period studied. Grey: starts as standard at the beginning but not standard at the end of the period studied. Yellow: up-to-standard for all periods studied.

**Figure 4 jcm-12-05947-f004:**
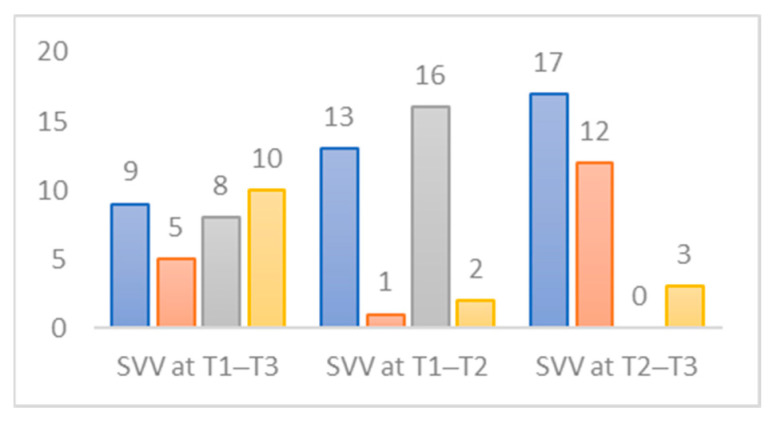
Change in normative status of the SVV indicator for three periods studied. Blue: not up-to-standard for all periods studied. Orange: not up-to-standard at the beginning but up-to-standard at the end of the period studied. Grey: starts as standard at the beginning but not standard at the end of the period studied. Yellow: up-to-standard for all periods studied.

**Figure 5 jcm-12-05947-f005:**
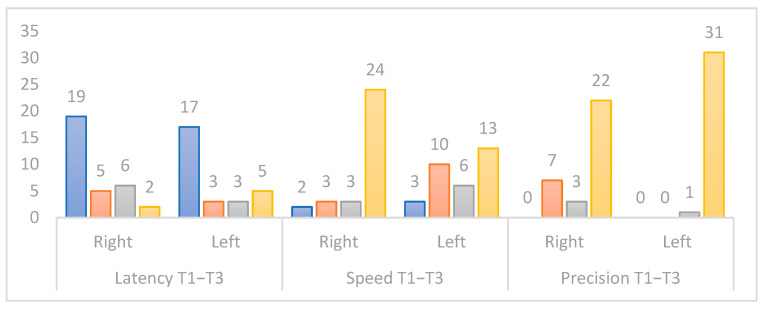
Change in normative status of the eye saccade indicator between the first and third periods studied. Blue: not up-to-standard for all periods studied. Orange: not up-to-standard at the beginning but up-to-standard at the end of the period studied. Grey: starts as standard at the beginning but not standard at the end of the period studied. Yellow: up-to-standard for all periods studied.

**Figure 6 jcm-12-05947-f006:**
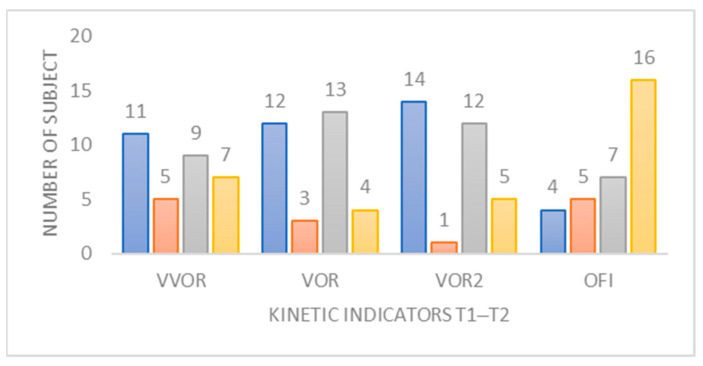
Change in normative status of the VVOR, VOR, IFO, and VOR2 indicator between the first and third periods studied. Blue: not up-to-standard for all periods studied. Orange: not up-to-standard at the beginning, but up-to-standard at the end of the period studied. Grey: starts as standard at the beginning but not standard at the end of the period studied. Yellow: up-to-standard for all periods studied.

**Figure 7 jcm-12-05947-f007:**
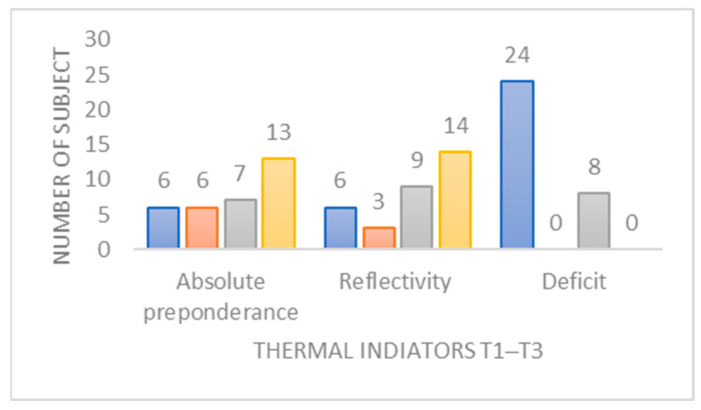
Change in normative status of the eye thermal indicators between the first and third periods studied. Blue: not up-to-standard for all periods studied. Orange: not up-to-standard at the beginning, but up-to-standard at the end of the period studied. Grey: starts as standard at the beginning but not standard at the end of the period studied. Yellow: up-to-standard for all periods studied.

**Table 1 jcm-12-05947-t001:** Description of the study population (Number = 32) operated on for vestibular Schwannoma at the hospital “Conception” (Marseille) between 1 January 2014 and 30 June 2019.

Variables	Measure Unit	Value	Percentage (%)
Sex	n		
Female		19	59.4
Male		13	40.6
Surgical treatment age	μ; s	59; 10.9	
Stage of tumor	n		
IIb		22	68.8
III		10	31.2
Size of tumor before surgery (mm)	μ; s	18; 5.4	
Side of pathology	n		
Right		17	53.1
Left		15	46.9
Preoperative cophosis	n	4	12.5
Preoperative vestibular physiotherapy	n	30	93.8
Preoperative visual dependence (VD)	n	21	65.6
Surgical approach	n		
RS		14	43.8
TL		18	56.2
Postoperative cophosis + 1 month	n	23	71.9
Postoperative vestibular physiotherapy	n	32	100

Legend: μ: mean; s: standard deviation; n: number, tumor stage: Kanzaki classification (Appendix A), RS: retro-sigmoid, TL: trans-labyrinthine.

**Table 2 jcm-12-05947-t002:** Rate of Variation of Postural Indicators Over the Three Studied Periods.

Indicators	T1–T3 (%)	*p*-Value	T1–T2 (%)	*p*-Value	T2–T3 (%)	*p*-Value
stable EO PII	5.2	0.86	10.2	0.110	−2.0	0.150
stable EC PII	−2.9	0.097 *	10.9	0.597	−7.4	0.029 **
stable OKN PII	−7.1	<0.01 ***	10.4	0.377	−16.4	<0.01 ***
unstable EO PII	11.49	0.86	47.0	0.014 **	−18.0	0.020 **
unstable EC PII	−7.22	0.076 *		0.856	−10.0	0.110
unstable OKN PII	−7.96	<0.01 ***		0.856	−10.0	0.043 **
Fall		0.45		<0.01 ***		<0.01 ***
VDI	−40.8	<0.01 ***	23.9	0.503	−42.2	<0.01 ***
SVV	843.9	0.2	1320.0	<0.01 ***	−32.6	<0.01 ***

Key interpretation: In T1–T3, no significant variation is observed despite a rate of variation of 5.2% (*p* = 0.86) for the “Stable YO PII” indicator. T1: one day before surgery. T2: seven days after surgery. T3: one month after surgery. PII: Postural instability index. Stable: stable ground condition. Unstable: unstable ground condition. EO: eyes open. EC: eyes closed. OKN: optokinetic nystagmus. VDI: Visual Dependency Index. SVV: Subjective Visual Vertical. Indicators with clinically positive evolution are in green, those with clinically negative evolution are in red, and those with no clinical interpretation or non-significant variation are in grey. In black: calculation not allowed by the analysis of rate variation. * trend towards significance, ** moderate significance; *** strong significance. Fall: a patient is considered a “faller” when they are unable to maintain their balance under physiological conditions of adaptive postural adjustments. Since the variable is binary, only the number of faller patients can be determined for each period.

**Table 3 jcm-12-05947-t003:** Rate of variation of VOG indicators during the three studied periods.

Indicators	T1–T3 (%)	*p*-Value	T1–T2 (%)	*p*-Value	T2–T3 (%)	*p*-Value
Right latency	−0.5	>0.99	−3.6	0.1	4.3	0.377
Left latency	−5.3	0.071	−8.4	0.011 **	8.3	0.377
Right velocity	−1.2	>0.99	−10.0	<0.01 ***	11.3	<0.01 ***
Left velocity	4.5	>0.99	−3.5	0.050 *	11.0	0.150
Right precision	−5.3	0.061 *	−7.3	<0.01 ***	2.8	0.215
Left precision	−0.3	0.572	−5.0	0.050 *	5.8	<0.01 ***

Key interpretation: in the T1–T3 period, no significant variation is observed despite a rate of variation of −0.5% (*p* > 0.99) for the “right calibration” indicator. T1: −1 day before surgery. T2: 7 days after surgery. T3: 1 month after surgery. VOG: video-oculography. Indicators with clinically positive evolution are in green, those with clinically negative evolution are in red, and those with no clinically interpretable evolution or non-significant variation are in grey. * trend towards significance, ** moderate significance; *** strong significance.

**Table 4 jcm-12-05947-t004:** Rate of variation of VNG_k_ indicators during the three studied periods.

Indicators	T1–T3 (%)	*p*-Value	T1–T2 (%)	*p*-Value	T2–T3 (%)	*p*-Value
VVOR gain	1.01	0.99	−5.0	0.042 **	16.0	0.179
VOR gain	67.34	0.99	82.0	<0.01 ***	18.0	0.785
VOR2 gain	−12.78	0.029 **	−13.0	0.071 *	4.0	0.680
IFO gain	223	0.458	51.0	> 0.99	241.0	0.458
VVOR prep	116	<0.01 ***	472.0	<0.01 ***	−55.0	<0.01 ***
VOR prep	225.64	0.377	826.0	<0.01 ***	−56.0	<0.01 ***
VOR2 prep	245.37	0.265	772.0	<0.01 ***	−55.0	<0.01 ***
IFO prep	306	0.15	224.0	0.701	109.0	0.362

Key interpretation: In the T1–T3 period, no significant variation is observed despite a rate of variation of 1.01% (*p* > 0.99) for the “VVOR gain” indicator. T1: 1 day before surgery. T2: 7 days after surgery. T3: 1 month after surgery. VNG_k_: Videonystagmography Kinetics. VVOR: Visuo-Vestibulo-Ocular Reflex. VOR: Vestibulo-Ocular Reflex. IFO: Index of Ocular Fixation. Prep: Preponderance. Indicators with clinically positive evolution are in green, those with clinically negative evolution are in red, and those with no clinically interpretable evolution or non-significant variation are in grey. * trend towards significance, ** moderate significance; *** strong significance.

**Table 5 jcm-12-05947-t005:** Rate of variation of VNG_t_ indicators during the three studied periods.

Indicators	T1–T3 (%)	*p*-Value	T1–T2 (%)	*p*-Value	T2–T3 (%)	*p*-Value
VNG_t_ Prep. Absolute	104.4	>0.99	276.4	<0.01 ***	−23.0	0.011 **
VNG_t_ Reflectivity	−9.3	0.11	−48.5	<0.01 ***	120.1	<0.01 ***
VNG_t_ Deficit	101.3	<0.01 ***	124.3	<0.01 ***	−7.7	0.017 **

Key interpretation: In T1–T3, no significant variation is observed despite a rate of change of 104.7% (*p* > 0.99) for the indicator “VNG_t_ Prep Absolute”. T1: 1 day before the operation. T2: 7 days after the operation. T3: 1 month after the operation. VNG: Videonystagmography. Prep: Preponderance. Indicators with clinically positive evolution are in green, those with clinically negative evolution are in red, and those with no clinically interpretable evolution or insignificant variation are in grey. ** moderate significance; *** strong significance.

**Table 6 jcm-12-05947-t006:** Rate of change of indicators in the dimensions of DHI, SF36, and PANQOL over the three studied periods.

Indicators	T1–T3 (%)	*p*-Value	T1–T2 (%)	*p*-Value	T2–T3 (%)	*p*-Value
SF36 Emotional Well-being	0.3	<0.01 ***	0.1	0.02 **	0.6	0.038 **
SF36 Social Functioning	0.08	0.824	0.1	0.454	0.0	>0.99
SF36 Fatigue/Energy	0.53	<0.01 ***	0.3	0.108	0.2	<0.01 ***
SF36 Physical Functioning	−1.98	0.99	1	>0.99	−2	>0.99
SF36 Physical Limitation	22.1	0.022 **	27	0.049 **	4	0.359
SF36 General Health	18.03	0.424	16	<0.01 ***	2	>0.99
PANQOL Hearing	0.05	0.585	−0.2	0.042 **	0.1	0.6
PANQOL Balance	145	0.281	105	>0.99	9	>0.99
PANQOL General Health	6.23	0.664	5	0.832	11	0.503
DHI Emotional		<0.01 ***		<0.01 ***		>0.99
DHI Physical	103	<0.01 ***	137	<0.01 ***	−22	0.019 **
DHI Functional	30.8	0.245	117	<0.01 ***	−52	<0.01 ***

Key interpretation: In T1–T3, a significant increase of 0.3% in the rate of change (*p* < 0.01) is observed for the indicator “SF 36 emotional well-being”. T1: −1 day before surgery. T2: 7 days after surgery. T3: 1 month after surgery. SF36: Short Form (36) health survey questionnaire. DHI: Dizziness Handicap Inventory questionnaire. PANQOL: Penn Acoustic Neuroma Quality-of-Life questionnaire. Indicators with clinically positive changes are shown in green, those with clinically negative changes are shown in red, and those with changes lacking clinical interpretation or with non-significant variations are shown in grey. In black: calculation not allowed by the analysis of rate variation. ** moderate significance; *** strong significance.

**Table 7 jcm-12-05947-t007:** Rate of change in 4PTA3 over the different periods.

Indicators	T1–T3 (%)	*p*-Value	T1–T2 (%)	*p*-Value	T2–T3 (%)	*p*-Value
4PTA3 Pathological side	109.7	<0.01 ***	101.0	<0.01 ***	5.3	0.508
4PTA3 Healthy side	−4.0	0.265	3.8	0.711	−5.2	0.013 **

Key: In T1–T3, a significant increase in the rate of change of 109.7% is observed: *p* < 0.01 for the 4PTA3 on the pathological side. T1: −1 day before surgery. T2: 7 days after surgery. T3: 1 month after surgery. 4PTA3: mean tonal hearing loss of airborne thresholds at 500, 1000, 2000, and 3000 Hz. Indicators with clinically positive changes are shown in green, those with clinically negative changes are shown in red, and those with changes lacking clinical interpretation or with non-significant variations are shown in grey. ** moderate significance; *** strong significance.

## Data Availability

The data presented in this study are available upon request from the corresponding author.

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
