# Peer review of "Identification of Follow-Up Markers for Rehabilitation Management in Patients with Vestibular Schwannoma"

_jcm, 2023, doi:10.3390/jcm12185947_

Round 1

Reviewer 1 Report

Thank you for inviting me to review this submission about the identification of monitoring markers for the rehabilitation of patients with vestibular schwannoma. Here are some comments/suggestions for the authors:

-       The abstract is hard to follow. The type of study is not clear in the abstracts, neither the methods or results. 

-       The syntaxis of the manuscript makes it difficult to follow the main idea, not only in the abstract but also throughout all the manuscript.

-       The introduction section is very extensive and distract the reader from the main goal of the study. It should be concise, 2 or 3 paragraphs only. Avoid using tables or images for the introduction section. 

-       In the methods section the name of the senior author is mentioned, mentioning direct names in the manuscript should be avoided always. It would be preferred “to a senior ENT consult”. Avoid terms like “D-1, D+7, +1 month”, please use terms like day-1, day-7 and 1-month. The inclusion criteria are duplicated, its described in the methods section and repeated in table 2. Please avoid the overuse of figures and tables if already mentioned through the manuscript. The methods section is extensive. A review of the literature has been made again as in the introduction section. I’d suggest to shorten both sections. I’d like to remark the authors effort to explain every single detail of their manuscript, however, it seems at some degree more like a narrative review rather than a original study.

-       The study is well structured; Unfortunately, I’d suggest authors to review this version of their study, try to simplify information and data collected, improve syntaxis to make it easier for readers to comprehend all data included. The research topic is interesting and novel. I’d be happy to re-read a new re submission of this study. 

many syntaxis errors all over the manuscript that makes it hard to follow the main idea. 

Author Response

Thank you for inviting me to review this submission about the identification of monitoring markers for the rehabilitation of patients with vestibular schwannoma. Here are some comments/suggestions for the authors:

# We thank the reviewer for his interest in this manuscript, and for his review and analysis.

-       The abstract is hard to follow. The type of study is not clear in the abstracts, neither the methods or results. The syntaxis of the manuscript makes it difficult to follow the main idea, not only in the abstract but also throughout all the manuscript.

# To make the manuscript easier to read, we have revised it entirely using the STROBE method.

-        The introduction section is very extensive and distract the reader from the main goal of the study. It should be concise, 2 or 3 paragraphs only. Avoid using tables or images for the introduction section. 

# To meet with the reviewer advice, we have shortened this section by ¼.

-       In the methods section the name of the senior author is mentioned, mentioning direct names in the manuscript should be avoided always. It would be preferred “to a senior ENT consult”. Avoid terms like “D-1, D+7, +1 month”, please use terms like day-1, day-7 and 1-month. The inclusion criteria are duplicated, its described in the methods section and repeated in table 2. Please avoid the overuse of figures and tables if already mentioned through the manuscript. The methods section is extensive. A review of the literature has been made again as in the introduction section. I’d suggest shortening both sections. I’d like to remark the authors effort to explain every single detail of their manuscript, however, it seems at some degree more like a narrative review rather than a original study.

# We have trimmed down these two sections according to the reviewer recommendations.

-       The study is well structured; Unfortunately, I’d suggest authors to review this version of their study, try to simplify information and data collected, improve syntaxis to make it easier for readers to comprehend all data included. The research topic is interesting and novel. I’d be happy to re-read a new re submission of this study. 

# We have chosen to omit a portion of the data, as it was too dense for the publication format. We hope that this will make the reading experience smoother and that the response to the primary objective is now more prominent. The secondary objective, which was presented in the initial version, will be the subject of a separate publication along with the remaining data.

Comments on the Quality of English Language

many syntaxis errors all over the manuscript that makes it hard to follow the main idea.

# We have completely revised our writing and added more rigor to the translation of sentence structures. We agree that some phrasings were giving a distorted meaning, sometimes contrary to the original idea.

Reviewer 2 Report

I thank the authors to have the opportunity to review their article "Identification of monitoring markers for rehabilitation management in patients with vestibular schwannoma" submitted to JCM.

Overall, the topic is highly relevant and very interesting. Being a surgeon that operates on vestibular schwannoma, identifying markers that help to predict vestibular rehabilitation in these patients, either after conservative management, after Radiosurgery, or after neurosurgical removal.

However, I have several issues with the manuscript in the current form:

- STROBE compliance: The abstract and the manuscript do not adhere to the  STrengthening the Reporting of OBservational studies in Epidemiology

- Abstract: The abstract does not reflect the cohort based original data analysis of 155 patients, but reads as a scoping review. This is not correct

- Amount: The manuscript is completely overloaded with information. It is highly confusing to read through the huge amount of information provided by the authors. it is only in the methods section where one understands that this is a original data cohort analysis of 155 VS patients.

- Tables : the structure of your presentation is highly confusing. Table 1 should be a baseline table of the analyzed cohort. Table 2 should be an outcome table of the analyzed cohort. All the other tables that describe analysis and methodology aspects should be cited within the discussion.

- Figures: The same accounts for the Figures. Figure 1 should be an in and -excludion flowchart highlighting all the aspect of when and which patients were in- and excluded from your database (STROBE). Figures describing other things should be cited in the Methods / discussion where appropriate

- Streamlining: the authors should significantly shorten their manuscript to streamline the content to the main question: which markers are predictive and relevant for VS patients in terms of vestibular rehabilitation (as outcome). The authors should account for "conservative treatment", "Radiosurgical treatment" or "neurosurgical treatment" and KOOS grading at presentation. The whole Abstract/ introduction / methodology / discussion /  conclusion / Tables / Figures need to be streamlines, shortened and focused to answers this questions that is reflected within the title. at least 50% of the content needs to be moved to the supplemental material section

Significant shortening of the manuscript with "short and concise medical scientific english language"

Author Response

I thank the authors to have the opportunity to review their article "Identification of monitoring markers for rehabilitation management in patients with vestibular schwannoma" submitted to JCM.

Overall, the topic is highly relevant and very interesting. Being a surgeon that operates on vestibular schwannoma, identifying markers that help to predict vestibular rehabilitation in these patients, either after conservative management, after Radiosurgery, or after neurosurgical removal.

However, I have several issues with the manuscript in the current form:

# We thank the reviewer for his thorough reading and analysis of the manuscript and for his interest in this work. Indeed, this article is intended for an audience of surgeons and physiotherapists, aiming to share the expertise of both professions in the post-operative management of patients with vestibular schwannoma.

- STROBE compliance: The abstract and the manuscript do not adhere to the Strengthening the reporting of observational studies in epidemiology

# We have revised the entire manuscript following the STROBE guidelines

- Abstract: The abstract does not reflect the cohort based original data analysis of 155 patients but reads as a scoping review. This is not correct.

# We have revised the abstract following the STROBE methodology

- Amount: The manuscript is completely overloaded with information. It is highly confusing to read through the huge amount of information provided by the authors. it is only in the methods section where one understands that this is a original data cohort analysis of 155 VS patients.

# We have rectified this aspect of the manuscript by incorporating this information in the introduction.

- Tables : the structure of your presentation is highly confusing. Table 1 should be a baseline table of the analyzed cohort. Table 2 should be an outcome table of the analyzed cohort. All the other tables that describe analysis and methodology aspects should be cited within the discussion.

# We presented the cohort in Table 1, as suggested. The results were subsequently streamlined for ease of comprehension: all results related to rate variations are presented in a table, which we condensed to simplify the presentation (color coding was added to enhance readability). Results based on standards were depicted in a more visually informative graphical format to highlight the temporal changes in this modality. We omitted half of the densely packed data to fit within the constraints of a single article, reserving the remainder for a future publication. Appendix A contains the tables with the collected data, while Appendix B includes only 2 supplementary tables that complement the interpretation of the results.

- Figures: The same accounts for the Figures. Figure 1 should be an in and -exclusion flowchart highlighting all the aspect of when and which patients were in- and excluded from your database (STROBE). Figures describing other things should be cited in the Methods / discussion where appropriate

# Figures have been modified as requested. Figure 1 was revised according to your advice, and optional figures have been removed.

- Streamlining: the authors should significantly shorten their manuscript to streamline the content to the main question: which markers are predictive and relevant for VS patients in terms of vestibular rehabilitation (as outcome). The authors should account for "conservative treatment", "Radiosurgical treatment" or "neurosurgical treatment" and KOOS grading at presentation. The whole Abstract/ introduction / methodology / discussion /  conclusion / Tables / Figures need to be streamlines, shortened and focused to answers this questions that is reflected within the title. at least 50% of the content needs to be moved to the supplemental material section

# To meet the reviewer advices, a streamlining of the data and their discussion has been carried out. We have omitted a portion of the data and plan to publish them in a follow up paper. Therefore, we hope that this version is simpler and more fluid in its reading.

Comments on the Quality of English Language

Significant shortening of the manuscript with "short and concise medical scientific english language"

# We shortened the manuscript, and tried to address the imperative of translation.

Round 2

Reviewer 1 Report

Thank you for inviting me to re-review your submission. This new version of the manuscript is entirely different and improved. I congratulate the authors for putting effort into improving all the recommendations suggested in the previous report. As mentioned before, this study is very interesting and has sufficient archivable value. All sections are better structured and easier to read in this improved manuscript. 

Reviewer 2 Report

The authors have followed meticulously my recommendations and significantly restructured and shortened the manuscript along scientific guidelines (STROBE) and recommendations.